# Survival and complications after neoadjuvant chemoradiotherapy versus neoadjuvant chemotherapy for esophageal squamous cell cancer: A meta-analysis

Yaru Guo[1,2‡], Mingna Xu[1,2], Yufei Lou[1,2], Yan Yuan[1,2], Yuling Wu[1,2], Longzhen Zhang[1], Yong Xin[1,2‡]*, Fengjuan Zhou[2,3‡]*

1 Department of Radiation, the Affiliated Hospital of Xuzhou Medical University, Xuzhou, Jiangsu, China, 2 First Clinical College, Xuzhou Medical University, Xuzhou, China, 3 Department of Radiation, the Second Affiliated Hospital of Xuzhou Medical University, Xuzhou, Jiangsu, China

‡ YX and FZ are co-corresponding authors. YG is first author of this manuscript.
* deep369@163.com (YX); 954811776@qq.com (FZ)

## Abstract

### Objectives

To compare the survival and complications of neoadjuvant chemoradiation (NCRT) versus neoadjuvant chemotherapy (NCT) for esophageal squamous cell carcinoma (ESCC).

### Methods

We conducted a systematic literature search of the PubMed, Web of Science, Cochrane Library, EMBASE, CNKI, Wanfang Data, CBM, and VIP databases from inception to November 2021. Meta-analyses were performed using RevMan (version 5.3) and Stata version 15.0.

### Results

A total of 18 studies were included, which involved 3137 patients, The results of the meta-analysis showed that the pathological complete remission rate (odds ratio [OR] = 5.21, 95% confidence interval [CI]: 2.85–9.50, p<0.00001) and complete tumor resection rate (OR = 2.31, 95% CI: 1.57–3.41, p<0.0001) in the NCRT group were significantly better than those in the NCT group. Our meta-analysis results showed that 1-, 3-, and 5-year survival rates (1-year overall survival [OS]: OR = 1.51, 95% CI: 1.11–2.05, p = 0.009; 3-year OS: OR = 1.73, 95% CI: 1.36–2.21, p<0.0001; 5-year OS: OR = 1.61, 95% CI: 1.30–1.99, p<0.00001) in the NCRT group were significantly higher than those in the NCT group. NCRT can lead a significant survival benefit compared with NCT and there was no significant difference between the two neoadjuvant treatments in terms of postoperative complications.

**Data Availability Statement:** All relevant data are within the manuscript and its Supporting information files.

**Funding:** This research is funded by Innovative Research Group Project of the National Natural Science Foundation of China. The funders had no role in study design, data collection and analysis, decision to publish, or preparation of the manuscript.

**Competing interests:** The authors have declared that no competing interests exist.

## Conclusion

The use of NCRT in the treatment of patients with ESCC patients showed significant advantages in terms of survival and safety relative to the use of NCT.

## Introduction

Esophageal cancer (EC) is the eighth most common cancer and sixth leading cause of cancer-related death worldwide [1, 2]. Esophageal squamous cell carcinoma (ESCC) is the main histological cancer type, accounting for approximately 80% of all ECs [3]. ESCC accounts for more than 90% of the total number of ECs in China [4].

Surgery has always been the main treatment for resectable locally advanced EC. However, the overall prognosis for esophagectomy alone is poor [5]. The 5-year overall survival rate after surgery is less than 15%, and the recurrence rate after radical resection of EC ranges 36.8–43.4%, with most recurrences occurring within 2 years after surgery [6–8]. Neoadjuvant treatment, with chemotherapy (NCT) or combined chemoradiotherapy (NCRT), followed by radical surgical resection is now the standard in curatively intended treatment. Neoadjuvant chemoradiation and neoadjuvant chemotherapy have been shown to improve survival in patients with locally advanced EC. According to the CROSS trial from the Netherlands and the NEOCRTEC5010 study from China,NCRT combined surgery can significantly improve the survival of patients with locally advanced EC [9]. The Japanese JCOG9907 study [10] and British OEO2 study [11] confirmed that NCT combined with surgery can bring significant survival benefits over those provided by surgery alone to patients with advanced EC. Neoadjuvant therapy (NCT/NCRT) plus surgery has been used as the standard treatment strategy for patients with locally advanced EC [12]. The Japanese guidelines favor NCT, while NCCN guidelines recommend NCRT as the first-line treatment option for advanced ESCC [13]. The superiority of either NCRT or NCT in the treatment of esophageal cancer remains controversial. Radiotherapy is expected to improve local control, whereas chemotherapy has the potential to eliminate micrometastases [14]. However, there are related studies that reported that neoadjuvant therapy itself is toxic and produces side effects, and that it may also increase the incidence of postoperative complications and mortality, adding radiation to NCT may exacerbate these side effects [15]. Therefore, there is no consensus on which neoadjuvant treatment is the most effective, and there is limited evidence comparing the long-term results of the two induction treatments. Accumulating evidence shows that Esophageal adenocarcinoma (EAC) and ESCC have different natural histories and different cellular origins; therefore, different pathological subtypes of EC have different sensitivities to chemotherapy and radiotherapy [16]. EAC and ESCC have different survival rates and prognoses [17]. ESCC appears to be more sensitive to tumor treatment than EAC, and patients with ESCC gained a greater survival benefit than patients with EAC in the CROSS trial [18].

We believe that these pathological subtypes should be analyzed separately when attempting to determine the best neoadjuvant treatment for esophageal cancer subtypes. Therefore, we collected all the experiments on NCT and NCRT for ESCC, including randomized controlled trials and retrospective experiments. This study aimed to systematically evaluate the survival and complications following NCRT and NCT for ESCC in meta-analysis and to provide evidence to guide the treatment of ESCC.

## Materials and methods

This protocol was registered with the International Platform of Registered Systematic Review and Meta-Analysis Protocols (INPLASY) on December 05, 2021, and was last updated on December 05, 2021 (registration number INPLASY2021120031).

**Study inclusion criteria**:

i.  Diagnosis of ESCC following cytological and histopathological examination in patients without serious cardiac, pulmonary, hepatic, or renal disease.

ii.  RCTs or Retrospective experiments comparing NCRT and NCT for treating ESCC.

iii.  The experimental design met the requirements and included patients with ESCC and EAC, and a subgroup analysis was performed with ESCC results reported separately.

iv.  The primary efficacy outcomes were pathological complete remission rate (pCR); complete (R0) tumor resection rate; 1-, 3-, and 5-year survival rates; toxicity of neoadjuvant treatment (including myelosuppression, gastrointestinal reaction, and esophagitis); and postoperative complications (including anastomotic leak, pulmonary complications, cardiac complications, chyle leak, and perioperative mortality).

**Study exclusion criteria**:

i.  The experimental design included ESCC and EAC were included, but the results for ESCC were not reported separately.

ii.  Studies reporting incomplete or inconsistent outcomes.

iii.  Duplicate studies, studies reporting animal experiments, case reports, cohort studies, and review articles.

## Search strategy and study selection

We identified all studies comparing NCRT and NCT in the treatment of EC in the PubMed, Cochrane Library, Web of Science, Embase, Wanfang Data, Chinese National Knowledge Infrastructure (CNKI), Chinese Biological Medicine (CBM) Database, and VIP Database published before November 2021. We used the following search terms: Esophageal Neoplasms, Neoadjuvant Chemoradiotherapy, and Neoadjuvant Chemotherapy. Two researchers (YG and YL) independently read the titles and abstracts of the identified papers, excluded irrelevant papers according to the inclusion and exclusion criteria. After the initial screening, the researchers read the full text to determine whether to include a study. When opinions were inconsistent, a third party judged whether to include the study.

## Data extraction and quality assessment

Two authors (MY and YY) independently extracted the relevant data, including authors; year of publication; country; number of cases; age; stage; NCT regimen; neoadjuvant radiotherapy dose; pCR rate; R0 resection rate; 1-, 3-, and 5-year overall survival (OS); toxicity of neoadjuvant treatment; and postoperative complications. The Cochrane Collaboration's tool was used to evaluate the quality of randomized studies, and the Newcastle-Ottawa quality assessment scale (NOS) was used for retrospective studies.

## Statistical analysis

All statistical analyses were performed using RevMan 5.3 and Stata 15.0. The results are presented as risk ratios (ORs) with 95% confidence intervals (CIs). The heterogeneity between studies were evaluated using Cochran's Q test and the $I^2$ statistic. A fixed-effects model was used if heterogeneity was not significant (p>0.1, $I^2<50.0\%$); otherwise, a random-effects model was applied. The results of the meta-analysis are presented as forest plots. Egger's test, and sensitivity analysis were used to evaluate publication bias. All p-values are two-sided, and a p-value less than 0.05 was considered statistically significant.

# Results

## Characteristics of studies

We identified 1326 studies in the initial database search and eliminated 417 duplicate articles and 241 reviews. We excluded 497 irrelevant research studies, 102 studies that studied NCRT combined with surgery vs surgery alone, 21 studies that studied CRT combined with surgery vs surgery alone, 5 studies that met the inclusion criteria that were in progress, 21 studies that included EAC and 4 studies that had inconsistent outcome indicators. Finally, 18 studies were included in the meta-analysis, of which seven [19–25] were randomized trials and eleven [26–36] were retrospective studies. The flow chart of the literature screening process is shown in Fig 1. A total of 2042 patients were included in the NCRT group, and 1095 patients were included in the NCT group. In the included studies, except for three retrospective studies [30, 31, 34] that did not detail the chemotherapy regimens and radiation doses, the chemotherapy regimens were either cisplatin plus 5-fluorouracil or cisplatin plus paclitaxel and the radiation doses were 30–45 Gy in the remaining included studies. The detailed characteristics of each included study are listed in Table 1.

## Quality assessment

The quality evaluation results of the seven RCT studies are shown in Fig 2. Three studies assigned random numbers, while none of the remaining studies described any particular randomization method. Only one study reported allocation concealment. Three studies that included patients who signed an informed consent did not follow a blinding method, and the remaining four studies did not have sufficient information to determine whether a blinding method was followed. All included studies presented complete data for analysis and no selective reporting or other biases were identified. The Newcastle–Ottawa scale was used to assess the quality of the eleven retrospective studies, and all scores were greater than 6 points; the evaluation results are shown in Table 1. This indicates that all the studies were of relatively high quality.

## pCR

Eleven included studies reported pCR [20, 22, 23, 26, 28–30, 32, 33, 35, 36]. There was heterogeneity between studies (p = 0.008, $I^2$ = 58%); therefore, a random-effects model was used for the meta-analysis. Meta-analysis showed that the pCR rate of the NCRT group was significantly higher than that of the NCT group (OR = 5.21, 95% CI:2.85–9.50, p<0.00001) (Fig 3).

## R0 resection

Ten included studies reported R0 resection [20, 22, 23, 25, 26, 28, 29, 32, 35, 36]. There was no significant heterogeneity between studies (p = 0.33, $I^2$ = 12%); therefore, the fixed-effects model was used for the meta-analysis. Meta-analysis showed that R0 resection in the NCRT

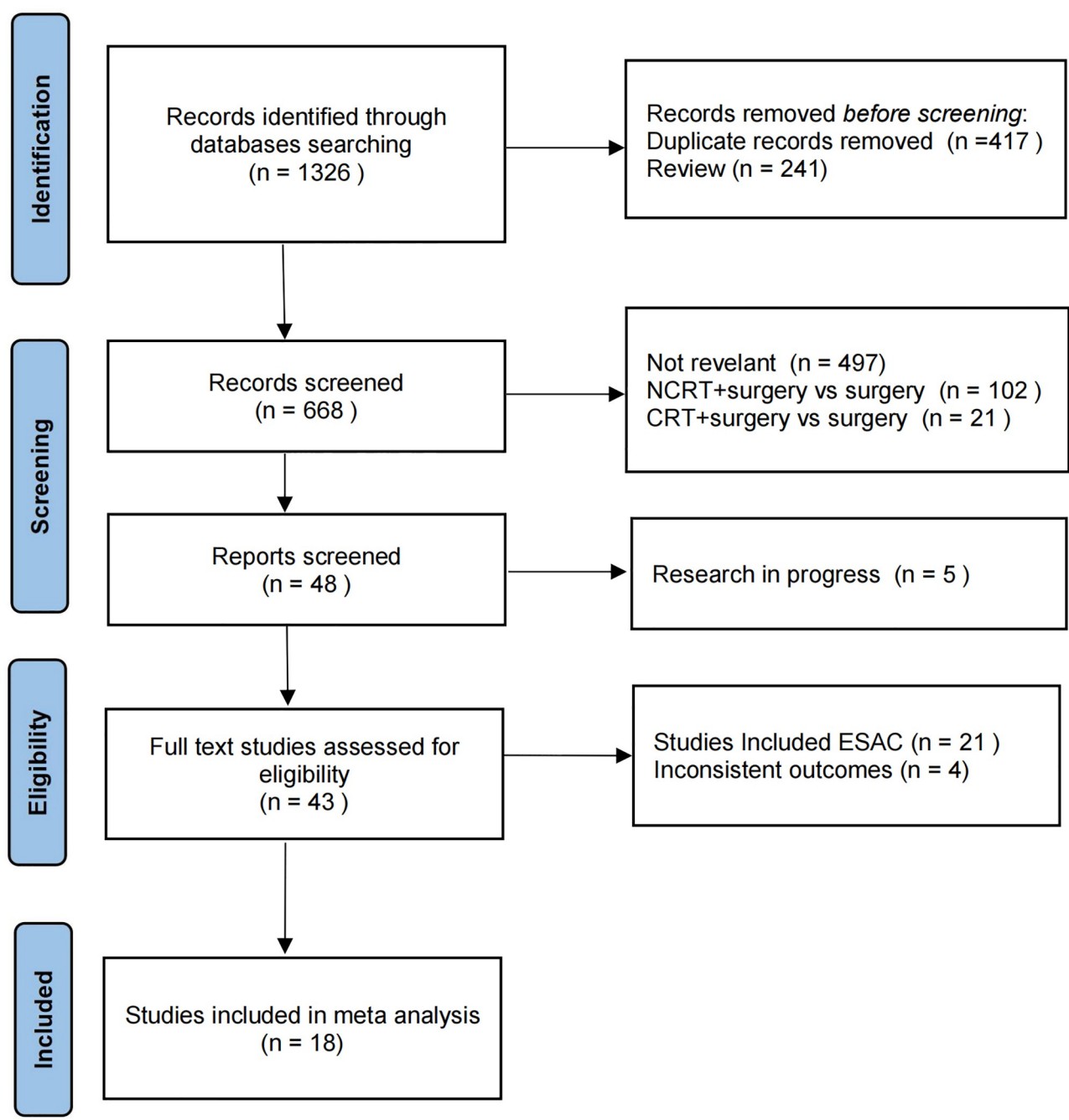

**Fig 1. Flow chart of studies screening.**

group was significantly higher than that in the NCT group (OR = 2.31, 95% CI:1.57–3.41, p<0.0001) (Fig 4).

## One-, three-, and five-year survival rates

Nine studies reported 1-year survival rates (p = 0.78, $I^2$ = 0%) and 3-year survival rates (p = 0.20, $I^2$ = 28%) [20–22, 25, 26, 28, 33, 35, 36], and the fixed-effects model was selected

**Table 1. The characteristics of studies included.**

| Author | Year | Country | Study design | n(nCRT/nCT) | Age/years | | Stage | nCRT | nCT | NOS |
|---|---|---|---|---|---|---|---|---|---|---|
| | | | | | nCRT | nCT | | | | |
| Qi WX [31] | 2021 | China | retrospective | 275/55 | 63±0.8 | 65±1.3 | II-IV | - | - | 6 |
| Zhang GC [29] | 2021 | China | retrospective | 90/90 | 18–75 | 18–75 | I-IVA | NDP/DDP+PTX/5Fu +32.4–50.0Gy | NDP/DDP+PTX/5-Fu | 6 |
| Wu YY [19] | 2020 | China | RCT | 40/40 | 58.4±7.5 | 59.2±8.3 | III-IV | PTX+DDP+41.4Gy | PTX+DDP | - |
| Fu LD [21] | 2019 | China | RCT | 60/60 | 57.6±9.5 | 56.5±9.8 | III-IV | PTX+DDP+40Gy | PTX+DDP | - |
| Miller [34] | 2018 | America | retrospective | 827/75 | 60.4±9.4 | 62.8±9.8 | I-IV | - | - | 7 |
| Nakashima [32] | 2018 | Japan | retrospective | 60/60 | 63.6 | 64.2 | II-IV | DDP+5-Fu+27-50Gy | DDP+5-Fu | 7 |
| Shi JR [27] | 2018 | China | retrospective | 46/46 | 55.05 ±15.28 | 56.05 ±15.86 | III-IV | DDP+5-Fu+55-66Gy | DDP+5-Fu | 8 |
| von Döbeln [23] | 2018 | Sweden | RCT | 25/25 | <70 | <70 | III-IV | DDP+5-Fu+40Gy | DDP+5-Fu | - |
| Hao DX [35] | 2017 | China | retrospective | 53/58 | 30–74 | 30–71 | II-III | DDP+5-Fu+40Gy | DDP+5-Fu | 7 |
| Tiesi [30] | 2017 | America | retrospective | 25/30 | 63±0.8 | 65±1.3 | II-IV | - | - | 7 |
| Li X [28] | 2017 | China | retrospective | 72/105 | 60(30–80) | 60(30–80) | II-III | DDP+5-Fu+36-40Gy | DDP+5-Fu | 8 |
| Zheng H [26] | 2017 | China | retrospective | 59/97 | 60(41–79) | 60(41–79) | III-IV | PTX+DDP+40Gy | PTX+DDP | 8 |
| Klevebro [36] | 2016 | Sweden | retrospective | 79/19 | 63 (20–80) | 63 (35–79) | I-III | DDP/ADM+5-Fu+40Gy | DDP/ADM+5-Fu | 7 |
| Skoczylas [24] | 2014 | Poland | RCT | 30/23 | - | - | II-III | DDP+5-Fu+30Gy | DDP+5-Fu | - |
| Lu J [20] | 2009 | China | RCT | 119/120 | 40–70 | 40–70 | II-IV | PTX+DDP+40Gy | PTX+DDP | - |
| Cao XF [22] | 2007 | China | RCT | 118/119 | - | - | II-IV | MMC+DDP+5-Fu+40Gy | MMC+DDP+5-Fu | - |
| Nakano [33] | 2001 | Japan | retrospective | 17/23 | 63.4±5.6 | 63.7±8.1 | II-IV | DDP+5-Fu+40Gy | DDP+5-Fu | 8 |
| Nygaard [25] | 1992 | Norway | RCT | 34/41 | 60.1(50–74) | 62.9(44–77) | I-III | BLM+DDP+35Gy | BLM+DDP | - |

because no significant heterogeneity was found. Ten included articles reported 5-year survival rates [20–22, 24, 26, 29, 31, 32, 34, 36], and the fixed-effects model was selected because significant heterogeneity was found (p = 0.10, $I^2$ = 39%). The results of the meta-analysis showed that the 1-year (OR = 1.51, 95% CI: 1.11–2.05, p = 0.009), 3-year (OR = 1.73, 95% CI: 1.36–2.21, p<0.0001), and 5-year survival rates (OR = 1.61, 95% CI: 1.30–1.99, p<0.00001) were higher in the NCRT group than in the NCT group. All differences were statistically significant (Fig 5).

## Postoperative complications

Ten studies reported perioperative mortality [22, 23, 25, 26, 28, 29, 32–35], ten studies reported anastomotic leak [20–22, 25, 26, 28, 29, 32, 33, 35], eight studies reported pulmonary complications [21, 25, 26, 28, 29, 32, 33, 35], five studies reported cardiac complications [21, 26, 28, 29, 35], four studies reported chyle leak [21, 22, 26, 35], four studies reported stricture [20, 22, 28, 35], three studies reported postoperative vocal cord paralysis [21, 26, 35], three studies reported postoperative bleeding [20, 22, 26], and two studies reported postoperative infection [26, 35]. No heterogeneity was found for any complication; therefore, a fixed-effects model was used for analysis. Table 2 shows the results of the meta-analysis of all postoperative complications. According to the results, the perioperative mortality rate of the NCRT group was similar to that of the NCT group (OR = 0.97, 95% CI: 0.55–1.72). The rates of anastomotic leak, pulmonary complications, cardiac complications, chyle leak, stricture, postoperative vocal cord paralysis, and infection in the NCRT group were higher than those in the CRT group, and the rate of bleeding in the NCRT group was lower than that in the CRT group;

**A**

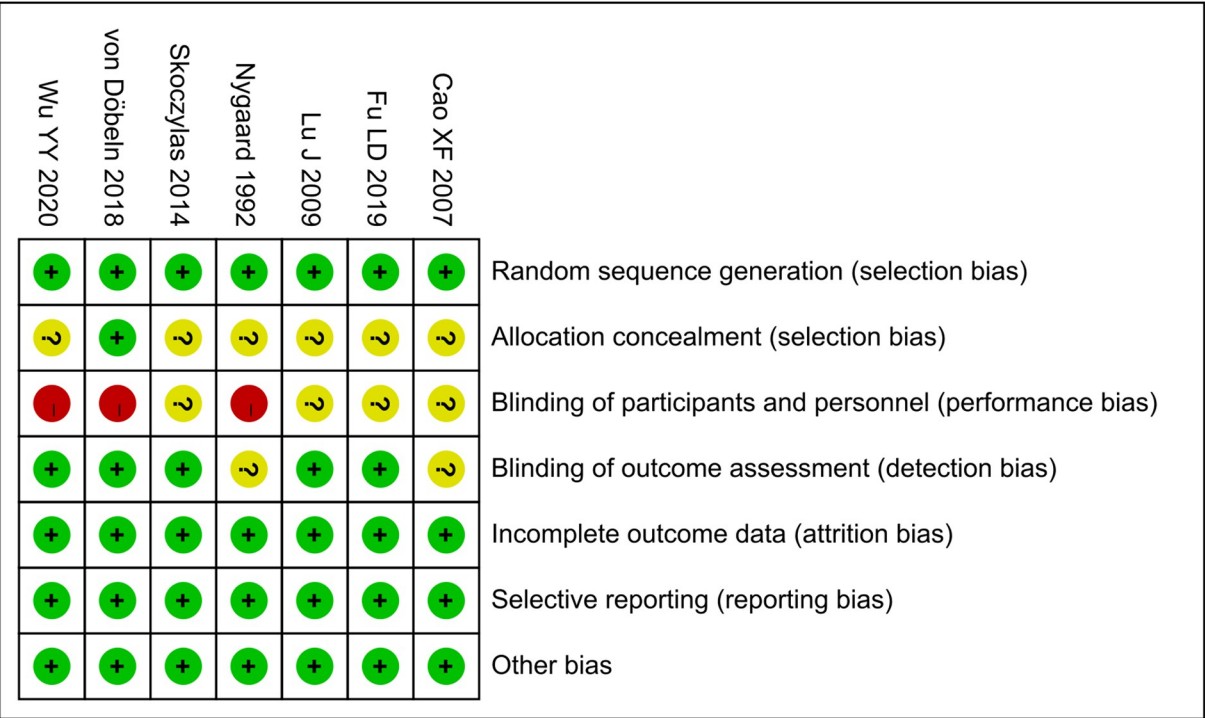

**B**

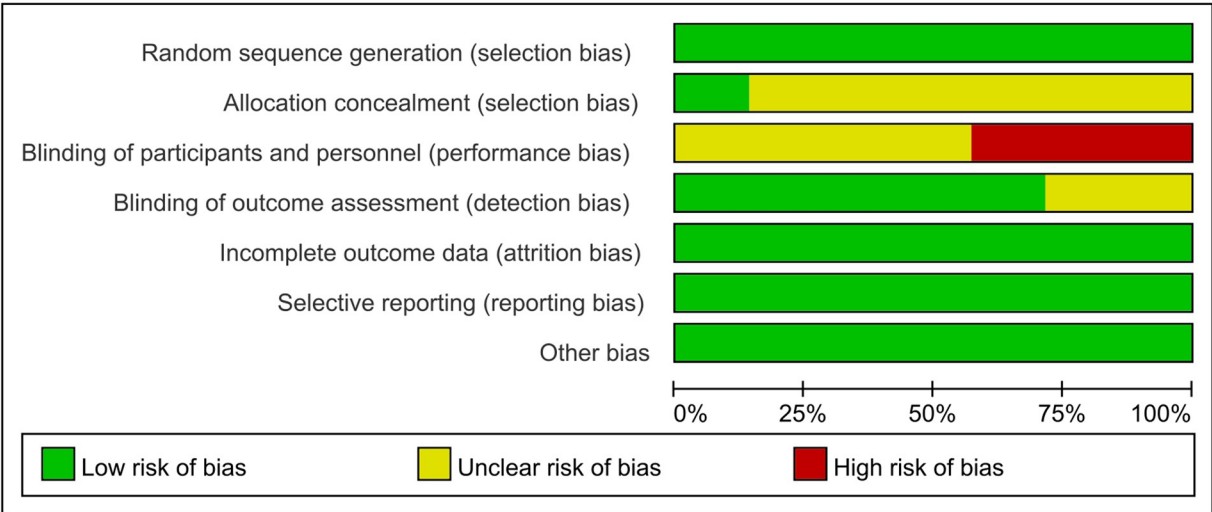

**Fig 2. Risk of bias summary (A) and bias graph (B).** Green indicates low risk; Red indicates high risk; Yellow indicates unknown risk.

however, The difference is not statistically significant (Fig 6). We considered no difference in complication rates between the two groups.

### Evaluation of sensitivity and publication bias

To ensure the accuracy and stability of the research, we conducted a sensitivity analysis by omitting each study to assess its effect on the overall results. Each study was not out of the

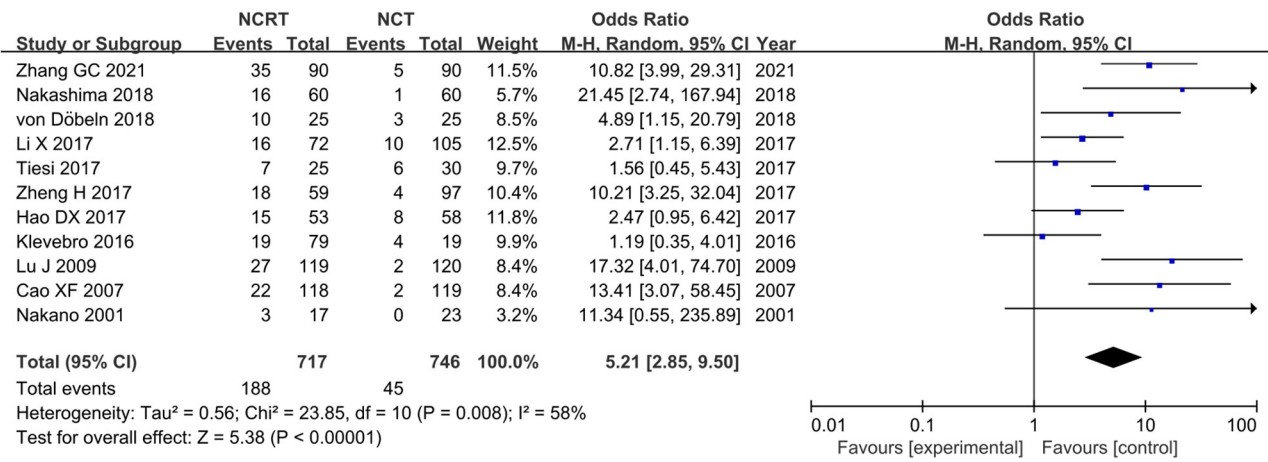

**Fig 3. Forest plot for pCR of NCRT group and NCT group.**

estimated ranges by visual inspection and had no significant effect on the results (Fig 7), indicating that the strong rationality and reliability of the results of our meta-analysis. We analyzed the symmetry of the funnel plot using Egger's regression test (S1 Fig), and the results showed that none of the articles had publication bias (Table 3), and all P-values > 0.05.

## Discussion

The incidence of EC is increasing, and EC is the the fourth most cause of cancer-related death in China [37]. In recent years, combined NCT and NCRT have become the standard treatment options for EC. However, the evidence to define the most beneficial neoadjuvant treatment for EC is insufficient [38]; consequently, which neoadjuvant regimen is most effective for patients with EC has remained a controversial topic. Patients with ESCC and ESAC all showed higher tumor responses to NCT and NCRT, but patients with ESCC showed higher sensitivity and significantly better survival after treatment with NCRT; in patients with ESAC, tumor response did not translate to improved survival. In addition to efficacy, NCT and NCRT treatment

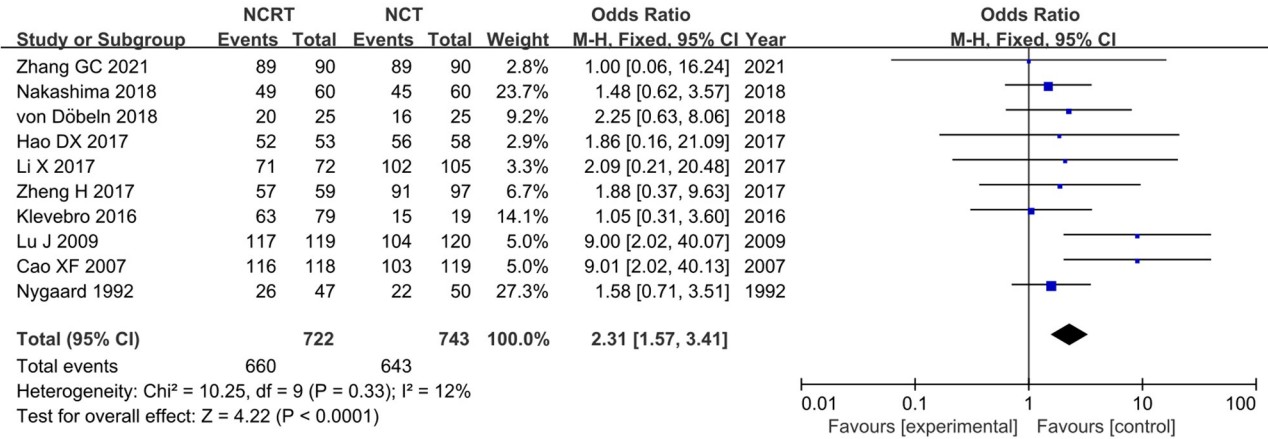

**Fig 4. Forest plot for R0 resection of NCRT group and NCT group.**

**A**

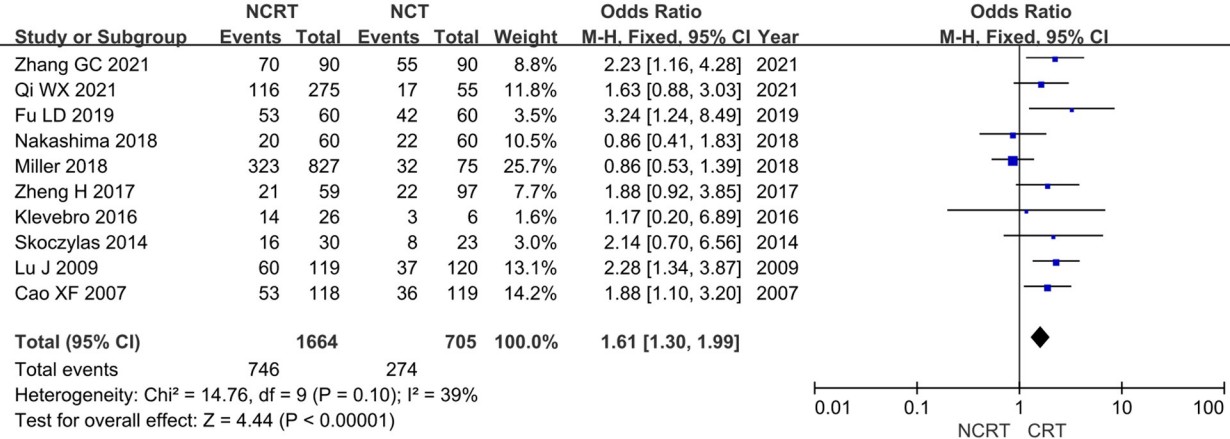

**B**

**C**

**Fig 5.** Forest plot for one-year (A), three-year (B), and five-year (C) survival rates of NCRT group and NCT group.

**Table 2. Summary of the results of meta-analysis.**

| Outcomes and complications | Included study | Events | Total of EXP | Events | Total of CON | Heterogeneiy | | Meta-analysis model | Result of meta-analysis | |
|---|---|---|---|---|---|---|---|---|---|---|
| | | | | | | P | I | | RR(95%CI) | P |
| pCR | 11 | 188 | 717 | 45 | 746 | 0.008 | 58% | random-effects | 5.21(2.85–9.50) | <0.00001 |
| R0 resection | 10 | 660 | 722 | 643 | 743 | 0.33 | 12% | fixed-effects | 2.31(1.57–3.41) | <0.0001 |
| 1-year OS | 9 | 517 | 618 | 509 | 651 | 0.78 | 0% | fixed-effects | 1.51(1.11–2.05) | 0.009 |
| 3-year OS | 9 | 373 | 597 | 324 | 645 | 0.20 | 28% | fixed-effects | 1.73(1.36–2.21) | <0.0001 |
| 5-year OS | 10 | 746 | 1664 | 274 | 705 | 0.10 | 39% | fixed-effects | 1.61(1.30–1.99) | <0.00001 |
| Perioperative mortality | 10 | 59 | 1355 | 23 | 693 | 0.85 | 0% | fixed-effects | 0.97(0.55–1.72) | 0.92 |
| Anastomotic leak | 10 | 62 | 682 | 60 | 773 | 0.11 | 37% | fixed-effects | 1.23(0.84–1.81) | 0.28 |
| pulmonary complications | 8 | 60 | 445 | 58 | 534 | 0.30 | 17% | fixed-effects | 1.31(0.88–2.13) | 0.18 |
| cardiac complications | 5 | 11 | 334 | 9 | 410 | 0.81 | 0% | fixed-effects | 1.44(0.60–3.47) | 0.42 |
| chyle leak | 4 | 6 | 290 | 6 | 334 | 0.69 | 0% | fixed-effects | 1.10(0.36–3.31) | 0.87 |
| stricture | 4 | 9 | 362 | 6 | 402 | 0.59 | 0% | fixed-effects | 1.66(0.62–4.48) | 0.31 |
| postoperative vocal cord paralysis | 3 | 8 | 172 | 7 | 215 | 0.54 | 0% | fixed-effects | 1.46(0.53–4.02) | 0.47 |
| bleeding | 3 | 2 | 296 | 3 | 336 | 0.95 | 0% | fixed-effects | 0.84(0.16–4.38) | 0.83 |
| Infection | 2 | 3 | 112 | 2 | 155 | 0.37 | 0% | fixed-effects | 1.79(0.34–9.58) | 0.49 |

options also need to consider the risk of toxicity and complications [23, 9, 39]. To ensure the completeness and reliability of the results, we included all RCTs and retrospective analyses of ESCC treatment published before November 2021.

The pCR is defined as fibrosis with or without inflammation extending through different layers of the esophageal wall, but with no viable residual tumor cells. Brücher et al. investigated the association between histopathological regression and survival in 311 patients with ESCC treated with NCRT/CRT. The results revealed that the histopathologic response classification according to the percentage of residual tumor cells was an independent prognostic factor (p<0.001). Non-responders had higher postoperative pulmonary morbidity, a higher 30-day mortality rate, and a lower survival rate compared to those of histopathologic responders (p<0.001). pCR was an independent significant predictor for EC, and patients who achieved pCR had better progression free survival(PFS) and OS than those without pCR [40]. The results of this meta-analysis showed that NCRT can result in better pCR rates in patients with ESCC than those achieved with CRT (OR = 5.21, 95% CI:2.85–9.50, p<0.00001), which means that it can provide better survival benefits.

Complete (R0) tumor resection was defined as complete tumor excision with all margins histologically free of tumor. R0 resection is the most important predictor of OS in patients with EC [11]. Both NCRT and NCT can increase the R0 resection rate of EC, and the R0 resection rate was higher in the NCRT group than in the surgery alone group in the CROSS study (92% vs. 65%). In the JCOG9907 study, R0 resection of the NCT group and surgery alone group was 85.4% and 65%, respectively. The results of our meta-analysis show that the R0 resection rate was higher in the NCRT group than in the NCT group (OR = 2.31, 95% CI:1.57–3.41, p<0.0001).

Neoadjuvant treatments can improve the survival rate of patients with ESCC. NCRT and NCT have obvious advantages compared to surgery alone; however, it is still unclear which of the two can provide greater survival benefits to patients with ESCC. In the literature that we included, except for 1-year OS of Daxuan Hao's study and 5-year OS of Nakashima and Miller's study were higher in the NCT group than in the NCRT group. The 1-year, 3-year, and 5-year OS rates in the NCRT group were higher than those in the NCT group in the remaining studies. Our meta-analysis showed that the 1-, 3-, and 5-year survival rates in the NCRT group were significantly better than those in the NCT group, The results were statistically significant.

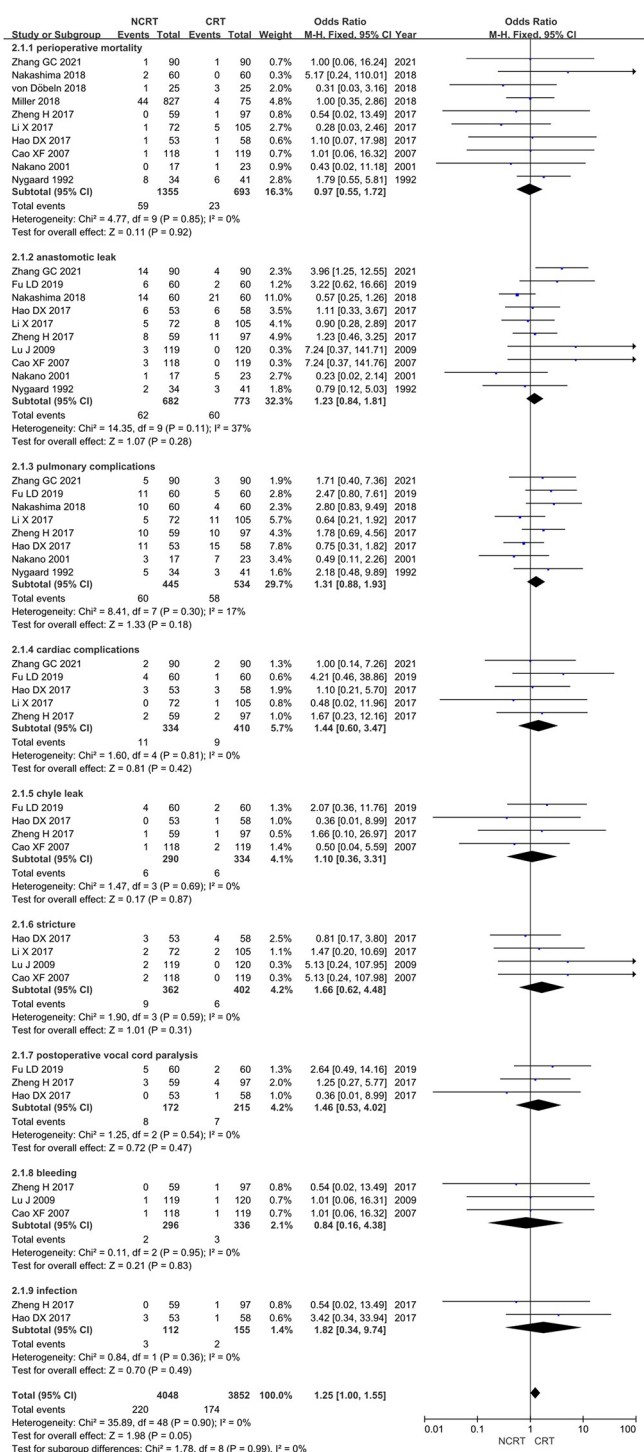

**Fig 6. Forest plot for postoperative complications of NCRT group and NCT group.**

A

B

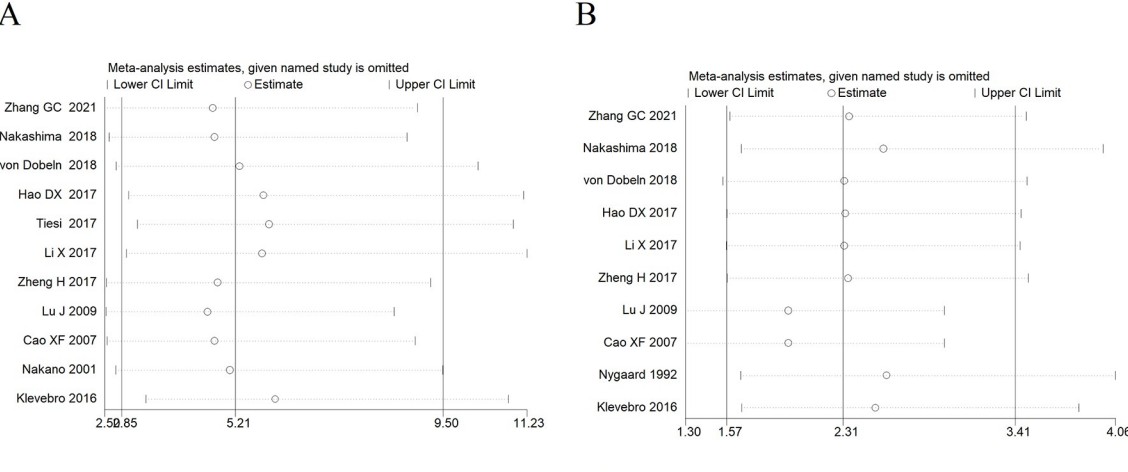

C

D

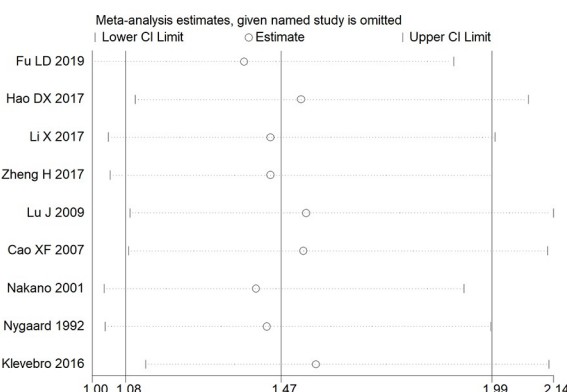

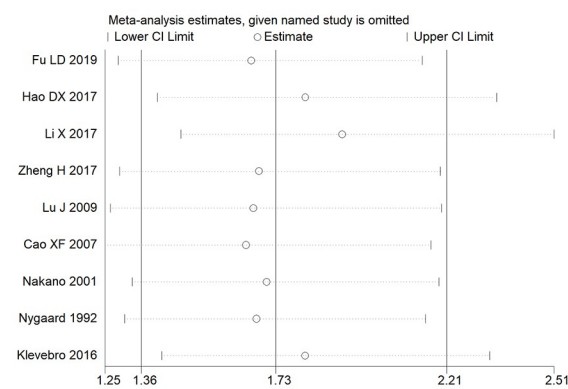

E

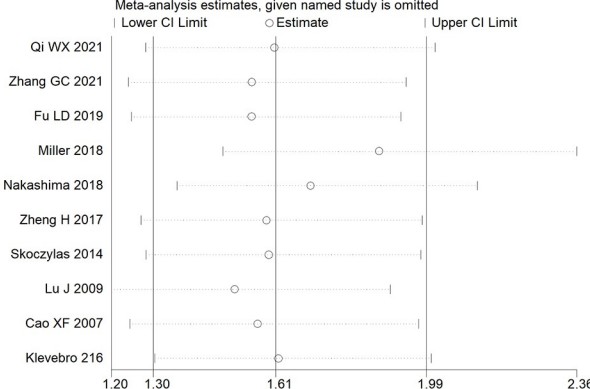

**Fig 7.** Sensitivity analysis of pCR (A), R0 resection (B), 1-year survival rates (C), 3-year survival rates (D) and 5-year survival rates (E) of NCRT group and NCT group.

**Table 3. Publication bias results of the included studies.**

| Evaluation Items | t | 95%CI | | P |
|---|---|---|---|---|
| pCR | 1.34 | -1.419 | 5.547 | 0.213 |
| R0 reaction | 0.79 | -1.401 | 2.868 | 0.451 |
| One-year OS | 0.30 | -2.769 | 3.58 | 0.771 |
| Three-year OS | 0.30 | -2.587 | 3.326 | 0.776 |
| Five-year OS | 0.60 | -2.348 | 3.985 | 0.568 |

NCT plus radiotherapy may increase treatment-related toxicity and postoperative complication rates [41]. In our meta-analysis, we also analyzed complications after neoadjuvant treatment, and the results showed that there was no significant difference in the incidence of myelosuppression, gastrointestinal reaction, or esophagitis between the two groups (S2 Fig). The results of our meta-analysis showed that there was no significant difference between the two neoadjuvant treatments in terms of postoperative complications and perioperative mortality. There is an ongoing trial CMISG1701 [42], which is a multicenter prospective randomized phase III clinical trial, comparing NCRT combined with minimally invasive surgery to NCT combined with minimally invasive surgery in patients with locally advanced resectable ESCC (cT3-4aN0-1M0) in China. Progress in minimally invasive surgery is bound to reduce postoperative complication rates. We look forward to the results of this study that may provide greater survival benefits to patients with ESCC.

In the treatment of ESCC, many scholars have realized that whether patients benefit from neoadjuvant therapy may be related to differences in gene expression, and people with certain molecular markers may be sensitive to neoadjuvant therapy [43]. Schneider et al. showed that the expression levels of thymidylate synthase (*TS*), dihydropyrimidine dehydrogenase (*DPD*), excision repair cross complementing factor 1 (*ERCC1*), glutathione S-transferase Pi (*GST-Pi*), epidermal growth factor receptor (*EGFR*), and *HER2* genes in ESCC resection specimens dropped significantly after neoadjuvant therapy [44]. However, there is no conclusive evidence that certain genes can guide NAT. Four studies of our included articles compared the complete clinical remission (cCR) rate after two neoadjuvant treatment. The results showed that the cCR of the NCRT group was higher than that of the NCT group (OR = 4.00, 95% CI: 2.41–6.64, p<0.00001) (S3 Fig). Two European trials, ESOSTRATE [45] and SANO [46], tested the hypothesis that patients with cCR after neoadjuvant therapy do not require esophagectomy. Patients receiving NCRT will be randomly divided into surgery and watchful waiting groups, and the main study parameter is overall survival. We look forward to the results of these two studies, which can show that patients who achieve complete remission after neoadjuvant treatment without surgery can achieve greater survival benefits and improved quality of life. Currently, immune-based treatment methods have shown great potential in the treatment of EC, and the combined treatment with NCT or NCRT regimens is more effective. CheckMate 577 was aimed at evaluating a checkpoint inhibitor as an adjuvant therapy in patients with resected EC or gastroesophageal junction cancer who had received NCRT. The results showed that the DFS of patients receiving adjuvant nivolumab was significantly greater than that of patients receiving placebo [47]. In the era of precision medicine, the treatment of EC should depend on the molecular biological characteristics of tumors and follow the principles of evidence-based medicine to formulate individualized treatment plans in a targeted manner, which will be the main direction of adjuvant radiotherapy and chemotherapy for ESCC in the future.

## Limitations

Our meta-analysis had some limitations. First, there were three studies where we could not obtain information on the neoadjuvant therapy. Second, because randomized trials directly comparing NCRT and NCT used in the treatment of EACC are limited, we also included prospective studies; as far as we know, there are three ongoing randomized controlled trials comparing NCRT vs NCT for patients with ESCC. The aim of JCOG1109 [48] which comes from Japan, was to confirm that docetaxel, cisplatin, and 5-fluorouracil are better than cisplatin plus 5-fluorouracil, and cisplatin plus 5-fluorouracil combined with chemotherapy and radiotherapy is better than cisplatin plus 5-fluorouracil as a preoperative treatment for ESCC. CMISG1701 [43] and HCHTOG1903 [49] are from China. HCHTOG1903 aimed to compare the OS rates of patients with locally advanced ESCC who received NCT and standard CROSS. We look forward to the results of these large randomized controlled clinical studies, which can guide us to better treat ESCC.

## Conclusion

In conclusion, compared with NCT in the treatment of ESCC, NCRT can can significantly increased the rates of pCR and R0 resection, and can more significantly improve the long-term survival of patients with ESCC without significantly increasing postoperative complications.

## Supporting information

**S1 Fig.** Egger's tests for publication bias of pCR (A), R0 resection (B), 1-year survival rates (C), 3-year survival rates (D) and 5-year survival rates (E) of NCRT group and NCT group. (TIF)

**S2 Fig.** Forest plot for myelosuppression (A), gastrointestinal reaction (B), esophagitis (C) between NCRT group and NCT group. (TIF)

**S3 Fig. Forest plot for cCR of NCRT group and NCT group.** (TIF)

**S1 Checklist. PRISMA checklist.** (DOC)

**S2 Checklist. PRISMA-P checklist.** (DOC)

## Author Contributions

**Data curation:** Yaru Guo, Mingna Xu, Yufei Lou, Yan Yuan.

**Formal analysis:** Yaru Guo, Yufei Lou, Fengjuan Zhou.

**Funding acquisition:** Yaru Guo, Longzhen Zhang.

**Investigation:** Yan Yuan, Yuling Wu.

**Project administration:** Yaru Guo.

**Resources:** Yaru Guo.

**Software:** Yaru Guo, Yufei Lou.

**Supervision:** Yong Xin, Fengjuan Zhou.

**Writing – original draft:** Yaru Guo.

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
