## [Decision Letter · Decision Letter 0]

18 Mar 2022

PONE-D-22-02701

Survival and complications after Neoadjuvant Chemoradiotherapy versus Neoadjuvant Chemotherapy for Esophageal Squamous Cell Cancer:a meta-analysis

PLOS ONE

Dear Dr. Zhou,

Thank you for submitting your manuscript to PLOS ONE. After careful consideration, we feel that it has merit but does not fully meet PLOS ONE’s publication criteria as it currently stands. Therefore, we invite you to submit a revised version of the manuscript that addresses the points raised during the review process.

We look forward to receiving your revised manuscript.

Kind regards,

Mona Pathak, PhD

Academic Editor

PLOS ONE

“This work was supported by National Natural Science Foundation of China (Grant No. 81972845).”

4. PLOS requires an ORCID iD for the corresponding author in Editorial Manager on papers submitted after December 6th, 2016. Please ensure that you have an ORCID iD and that it is validated in Editorial Manager. To do this, go to ‘Update my Information’ (in the upper left-hand corner of the main menu), and click on the Fetch/Validate link next to the ORCID field. This will take you to the ORCID site and allow you to create a new iD or authenticate a pre-existing iD in Editorial Manager. Please see the following video for instructions on linking an ORCID iD to your Editorial Manager account: https://www.youtube.com/watch?v=_xcclfuvtxQ.

Reviewers' comments:

Reviewer's Responses to Questions

**Comments to the Author**

1. Is the manuscript technically sound, and do the data support the conclusions?

Reviewer #1: Yes

Reviewer #2: Yes

2. Has the statistical analysis been performed appropriately and rigorously? 

Reviewer #1: Yes

Reviewer #2: Yes

3. Have the authors made all data underlying the findings in their manuscript fully available?

Reviewer #1: Yes

Reviewer #2: Yes

4. Is the manuscript presented in an intelligible fashion and written in standard English?

Reviewer #1: Yes

Reviewer #2: Yes

5. Review Comments to the Author

Reviewer #1: Dear author;

The article is well designed. Its constraints were adequately explained. Although the article's meta-analysis and statistics are an important feature, there are many publications from China on this subject. The subject is not original, I leave the comment here to the editor.

Reviewer #2: In this meta-analysis, the authors compared Neoadjuvant Chemoradiotherapy and Neoadjuvant Chemotherapy for Esophageal Squamous Cell Cancer and concluded the superiority of Neoadjuvant Chemoradiotherapy towards survival. This is an interesting and useful one of its kind study. Please address the following points; the 3rd and 4th points are important:

-The authors mentioned- 'After the initial screening, the researchers read the full text to determine whether to include a study'. Apart from the inclusion/exclusion criteria the authors enlisted, which others factors determined whether to include the study or not?

-Please mention a couple of lines about the 'allocation concealment' the authors mentioned.

-Please consider the drawing of Kaplan-Meier curves to show the survival difference between NCRT and NCT.

-Why did the authors not correlate the Clinicopathological features and survival outcomes of patients with NCRT/NCT? This analysis could potentially add much additional value to the manuscript.

6. PLOS authors have the option to publish the peer review history of their article (what does this mean?). If published, this will include your full peer review and any attached files.

Reviewer #1: No

Reviewer #2: No

---

## [Author Response · Author response to Decision Letter 0]

30 Apr 2022

Thank you very much to the two reviewers for their review of our paper and for their valuable comments. I am very grateful to the reviewers for their contributions to our paper. Below is our response to reviewers comments

Reply review one

Thank you for your encouragement. Although there have been related studies in China, most of them are not included in all the literature. This time, we searched all Chinese and English databases since the establishment of the database. With the literature on neoadjuvant chemotherapy, I hope to draw a relatively considerable conclusion, and provide more objective evidence-based medical evidence for neoadjuvant radiotherapy and chemotherapy of esophageal squamous cell carcinoma. Thanks again for your dedication to our articles.

Reply review two

Thank you very much for your valuable comments to us, thank you for your efforts on our article, we will answer your questions below。

Question one：The authors mentioned- 'After the initial screening, the researchers read the full text to determine whether to include a study'. Apart from the inclusion/exclusion criteria the authors enlisted, which others factors determined whether to include the study or not?

My answer: Because we are ultimately comparing neoadjuvant chemoradiotherapy versus neoadjuvant chemotherapy for esophageal squamous cell carcinoma, there are many trials studying esophageal cancer (including esophageal adenocarcinoma and esophageal squamous cell carcinoma), so we need to read this kind of literature carefully , to see if it separately reported the efficacy of neoadjuvant chemoradiotherapy versus neoadjuvant chemotherapy for esophageal squamous cell carcinoma. If there is, we need to extract this part of the data, and this part of the data also meets our inclusion criteria.

Question two: Please mention a couple of lines about the 'allocation concealment' the authors mentioned.

My answer: Allocation hiding mainly refers to the randomness of whether the people who implement the allocation in the experiment strictly implement the results of random numbers.

Question three: Please consider the drawing of Kaplan-Meier curves to show the survival difference between NCRT and NCT.

My answer: Thank you for your suggestion, but I'm very sorry, because our research is a secondary research based on other people's published research. The survival curve we understand should be the survival curve based on one or several influencing factors in a study, but because our research is based on many published studies, currently due to our limited ability, there is no way to Survival curves were drawn based on the literature we included. We are very sorry, we feel your suggestion is good and we need to study further on your suggestion.

Question four: Why did the authors not correlate the Clinicopathological features and survival outcomes of patients with NCRT/NCT? This analysis could potentially add much additional value to the manuscript.

My answer: Thank you very much for your suggestion, we also think it is very good suggestion. As our answer to a question above, because our research is a secondary research based on the research that others have published. So what we want to do is very limited. We can only perform a statistical analysis of studies that have been reported in the included literature. Since all the literatures on the efficacy of neoadjuvant chemoradiotherapy and neoadjuvant chemotherapy for esophageal squamous cell carcinoma were included in our study, and there was no literature linking clinicopathological characteristics and survival outcomes, we had no way to complete this idea. I am very sorry, but I think your suggestion is very good, and it has great guiding significance for our future scientific research work. Thank you for your efforts in our article. Thank you again from the bottom of my heart.

---

## [Decision Letter · Decision Letter 1]

27 Jun 2022

Survival and complications after Neoadjuvant Chemoradiotherapy versus Neoadjuvant Chemotherapy for Esophageal Squamous Cell Cancer:a meta-analysis

PONE-D-22-02701R1

Dear Dr. Zhou,

We’re pleased to inform you that your manuscript has been judged scientifically suitable for publication and will be formally accepted for publication once it meets all outstanding technical requirements.

Kind regards,

Mona Pathak, PhD

Academic Editor

PLOS ONE

Additional Editor Comments (optional):

Reviewers' comments:

Reviewer's Responses to Questions

**Comments to the Author**

1. If the authors have adequately addressed your comments raised in a previous round of review and you feel that this manuscript is now acceptable for publication, you may indicate that here to bypass the “Comments to the Author” section, enter your conflict of interest statement in the “Confidential to Editor” section, and submit your "Accept" recommendation.

Reviewer #2: All comments have been addressed

2. Is the manuscript technically sound, and do the data support the conclusions?

Reviewer #2: Yes

3. Has the statistical analysis been performed appropriately and rigorously? 

Reviewer #2: Yes

4. Have the authors made all data underlying the findings in their manuscript fully available?

Reviewer #2: Yes

5. Is the manuscript presented in an intelligible fashion and written in standard English?

Reviewer #2: Yes

6. Review Comments to the Author

Reviewer #2: (No Response)

7. PLOS authors have the option to publish the peer review history of their article (what does this mean?). If published, this will include your full peer review and any attached files.

Reviewer #2: No

---

## [Editor Report · Acceptance letter]

30 Jun 2022

PONE-D-22-02701R1 

Survival and complications after Neoadjuvant Chemoradiotherapy versus Neoadjuvant Chemotherapy for Esophageal Squamous Cell Cancer：a meta-analysis 

Dear Dr. Zhou:

I'm pleased to inform you that your manuscript has been deemed suitable for publication in PLOS ONE. Congratulations! Your manuscript is now with our production department. 

Kind regards, 

on behalf of

Dr. Mona Pathak 

Academic Editor

PLOS ONE